# Spatial determination and prognostic impact of the fibroblast transcriptome in pancreatic ductal adenocarcinoma

Wayne Croft[1,2†], Hayden Pearce[1†], Sandra Margielewska-Davies[1], Lindsay Lim[3], Samantha M Nicol[1], Fouzia Zayou[1], Daniel Blakeway[1], Francesca Marcon[4], Sarah Powell-Brett[4], Brinder Mahon[4], Reena Merard[4], Jianmin Zuo[1], Gary Middleton[1,4], Keith Roberts[4], Rachel M Brown[4], Paul Moss[1,4*]

[1]Institute of Immunology and Immunotherapy, College of Medical and Dental Sciences, University of Birmingham, Birmingham, United Kingdom; [2]Centre for Computational Biology, University of Birmingham, Birmingham, United Kingdom; [3]Cancer Research Horizons, The Francis Crick Institute, London, United Kingdom; [4]University Hospitals Birmingham NHS Foundation Trust, Queen Elizabeth Hospital Birmingham, Birmingham, United Kingdom

*For correspondence:
p.moss@bham.ac.uk

†These authors contributed equally to this work

Competing interest: The authors declare that no competing interests exist.

**Abstract** Pancreatic ductal adenocarcinoma has a poor clinical outcome and responses to immunotherapy are suboptimal. Stromal fibroblasts are a dominant but heterogenous population within the tumor microenvironment and therapeutic targeting of stromal subsets may have therapeutic utility. Here, we combine spatial transcriptomics and scRNA-Seq datasets to define the transcriptome of tumor-proximal and tumor-distal cancer-associated fibroblasts (CAFs) and link this to clinical outcome. Tumor-proximal fibroblasts comprise large populations of myofibroblasts, strongly expressed podoplanin, and were enriched for Wnt ligand signaling. In contrast, inflammatory CAFs were dominant within tumor-distal subsets and expressed complement components and the Wnt-inhibitor SFRP2. Poor clinical outcome was correlated with elevated HIF-1α and podoplanin expression whilst expression of inflammatory and complement genes was predictive of extended survival. These findings demonstrate the extreme transcriptional heterogeneity of CAFs and its determination by apposition to tumor. Selective targeting of tumor-proximal subsets, potentially combined with HIF-1α inhibition and immune stimulation, may offer a multi-modal therapeutic approach for this disease.

## Editor's evaluation

The plasticity and heterogeneity of fibroblasts in the tumor microenvironment of pancreatic ductal adenocarcinoma (PDAC) has emerged as a key factor in determining tumor growth and therapeutic response. Here the authors use innovative approaches to combine spatial profiling with single cell transcriptomics to define tumor-proximal populations of fibroblasts that predict clinical outcome. Specifically, elevated expression of HIF-1a and podoplanin predicted worse outcome while inflammatory gene expression correlated with increased survival, suggesting future interventions targeting proximal fibroblast populations to mitigate against PDAC.

## Introduction

Therapeutic control of pancreatic ductal adenocarcinoma (PDAC) is one of the greatest challenges in oncology and PDAC remains associated with poor long-term survival (*Arnold et al., 2019*). Although

**eLife digest** Pancreatic cancer is one of the deadliest and most difficult cancers to treat. It responds poorly to immunotherapy for instance, despite this approach often succeeding in enlisting immune cells to fight tumours in other organs. This may be due, in part, to a type of cell called fibroblasts. Not only do these wrap pancreatic tumours in a dense, protective layer, they also foster complex relationships with the cancerous cells: some fibroblasts may fuel tumour growth, while other may help to contain its spread.

These different roles may be linked to spatial location, with fibroblasts adopting different profiles depending on their proximity with cancer calls. For example, certain fibroblasts close to the tumour resemble the myofibroblasts present in healing wounds, while those at the periphery show signs of being involved in inflammation. Being able to specifically eliminate pro-cancer fibroblasts requires a better understanding of the factors that shape the role of these cells, and how to identify them.

To examine this problem, Croft et al. relied on tumour samples obtained from pancreatic cancer patients. They mapped out the location of individual fibroblasts in the vicinity of the tumour and analysed their gene activity. These experiments helped to reveal the characteristics of different populations of fibroblasts. For example, they showed that the myofibroblast-like cells closest to the tumour exhibited signs of oxygen deprivation; they also produced podoplanin, a protein known to promote cancer progression. In contrast, cells further from the cancer produced more immune-related proteins.

Combining these data with information obtained from patients' clinical records, Croft et al. found that samples from individuals with worse survival outcomes often featured higher levels of podoplanin and hypoxia. Inflammatory markers, however, were more likely to be present in individuals with good outcomes.

Overall, these findings could help to develop ways to selectively target fibroblasts that support the growth of pancreatic cancer. Weakening these cells could in turn make the tumour accessible to immune cells, and more vulnerable to immunotherapies.

immunotherapy has transformed the clinical outlook for many tumor subtypes its impact on PDAC has been disappointing to date. One factor in this regard may be the characteristic nature of the PDAC microenvironment which is associated with an intense desmoplastic reaction characterized by an abundance of cancer associated fibroblasts (CAF; *Biffi and Tuveson, 2021*; *Menezes et al., 2022*).

Fibroblasts are key regulators of tumor biology and there is now intense interest in understanding how they may act to maintain or suppress tumor growth. CAF are the predominant source of extracellular matrix and develop a complex system of interactions with tumor cells. Although some of the characteristics of CAF suggest chronic activation with sustained production of alpha-small muscle actin(α-SMA), they differ from normal counterparts by relative resistance to apoptosis or reversion of quiescence (*Piersma et al., 2020*). PDAC-associated CAF can limit the access of immune effector cells to tumor (*Mhaidly and Mechta-Grigoriou, 2021*; *Inoue et al., 2016*; *Ene-Obong et al., 2013*), promote the infiltration of immune suppressive leucocyte populations and directly support tumor growth (*Kumar et al., 2017*; *Albrengues et al., 2014*; *Mezawa and Orimo, 2016*; *Fang et al., 2019*). However, studies from murine models have also shown that fibroblasts may also play a protective role in limiting tumor metastasis and that targeting of stromal cells can accelerate disease progression (*Özdemir et al., 2014*). Furthermore, a clinical trial that targeted CAF through inhibition of the hedgehog protein, combined with chemotherapy, was terminated early due to disease progression (*Catenacci et al., 2015*). One suggestion has been that fibroblasts may play an important role in constraining the growth of of early-stage tumors but subsequently become subverted to support tumor progression (*Menezes et al., 2022*). As such, effective fibroblast-targeted therapies will need to be directed selectively towards those subpopulations that are critical to support tumor growth.

Transcriptional analyses have shown at least four subsets of CAF in PDAC with a heterogenous profile and direct associations with clinical outcome (*Mezawa and Orimo, 2016*). Spatially related features are also observed with α-SMA[high] myofibroblast populations closely associated with tumor whilst inflammatory subsets reside more distally (*Öhlund et al., 2017*). Cytokines such as TGF and IL-1 appear as key regulators and may direct differentiation from a CD105 +precursor (*Dominguez et al., 2020*). Marked cellular plasticity is a feature of CAF (*Neuzillet et al., 2019*), although it remains unclear

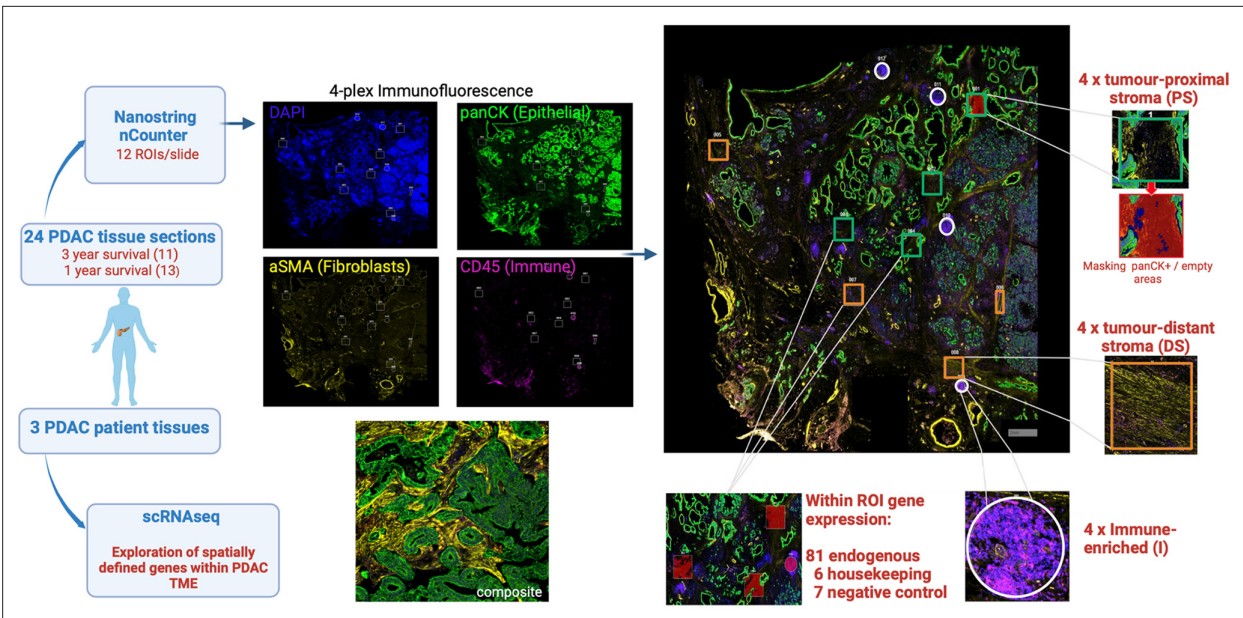

**Figure 1.** Schematic representation of experimental approach. Histological slides of 24 surgical resection specimens from patients with PDAC were stained with DAPI ('nuclear'), anti-pan-CK ('epithelial'), anti-α-SMA ('fibroblast') and anti-CD45 ('immune') to identify tumor cells and primarily to define three domains: 'tumor-proximal stroma' (PS), 'tumor-distant stroma' (DS) and 'immune-enriched' (I). The NanoString Immuno-oncology RNA probe set, in combination with a custom panel of 10 fibroblast-targeted RNA probes, was used to interrogate four areas (termed 'Regions of Interest'; ROI) from each of the three domains using the NanoString GeoMx Digital Spatial Profiler (DSP) platform. The transcriptional profile of spatially-defined tumor-proximal or tumor-distant fibroblast cells was subsequently aligned to scRNA-Seq datasets from three additional patients with PDAC.

if CAF develop discrete lineages or are interchangeable. Within PDAC a population of LRRC15+ myofi-broblasts is emerging as a potential tumor-associated subset with direct impact on clinical responses to therapy (*Dominguez et al., 2020*).

Immunotherapy trials in PDAC indicate that multi-modal approaches may be required for effective therapy and therapeutic targeting of CAF subpopulations could contribute to this. Here, we combine spatial transcriptomics with a scRNA-Seq dataset to define the transcriptome of tumor-proximal and tumor-distal fibroblasts within the PDAC microenvironment. Furthermore, these studies were correlated with clinical outcome and revealed that elevated podoplanin and HIF-1α expression were markers of poor outcome whilst expression of immunoregulatory genes correlates with favorable long-term response. These findings provide insight into stromal architecture in PDAC and could help to guide therapeutic approaches to target pro-tumorigenic fibroblast subsets.

## Results

### Spatially defined stromal and immune regions can be characterized within the PDAC microenvironment

Histological slide sections were obtained from tumor biopsies of 24 patients with pancreatic ductal adenocarcinoma (PDAC) who had undergone surgical resection for localized disease. Thirteen patients had died of PDAC within 12 months of diagnosis (subsequently referred to as 'poor response') whilst 11 had survived for at least 36 months ('good response').

Four-plex immunofluorescence staining was used initially to define major anatomical subregions of the tumor. Antibodies against pan-cytokeratin, α-SMA and CD45 identified epithelial, fibroblast and immune populations respectively whilst DAPI staining defined nuclear architecture. Regions in which stromal cells were adjacent to tumor ('tumor-proximal stroma'), distant from tumor ('tumor-distant stroma') or enriched for CD45+ immune cells ('immune enriched') were then selected and 4 areas ('regions of interest'; ROI) within each of these 3 domains were selected from each patient for assessment using the NanoString GeoMx Digital Spatial Profiler (DSP) platform (*Figure 1*).

Ninety-four RNA hybridization probes (*Supplementary file 1*, *Figure 2—figure supplements 1 and 2*) for 81 endogenous, 6 housekeeping and 7 negative control genes were then applied to the 12 ROI from each patient, thus generating 288 transcriptional datasets. Six pan-cytokeratin positive ROI were also selected to define the transcriptional profile of tumor cells.

Hierarchical clustering of transcriptional datasets delineated immune and stromal regions with two immune profiles clustering separately due to differential expression of activatory and inhibitory immune genes. Cell-type expression profiles were consistent with immune or stromal origin (*Figure 2A*). UMAP analysis broadly separated tumor, proximal stroma, distal stroma, and immune regions (*Figure 2B*), although overlay of clinical outcome data did not reveal significant clustering.

Expression of cell lineage marker genes was then used to determine the relative localization of cell subsets within proximal-stromal, distal-stromal, immune or tumor regions of interest (*Figure 2C*). These confirmed localization of epithelial, fibroblast and lymphoid cells within the tumor, stromal and immune regions respectively whilst monocyte representation was equivalent within stromal and immune regions, consistent with broad infiltration within PDAC microenvironment. CD3E and MS4A1 expression indicated that stromal regions also contained smaller populations of infiltrating T and B cells (*Figure 2C*). Distinct modules of genes co-expressed with lineage markers could also be identified and were consistent with cell type (*Figure 2—figure supplement 3*). Stromal regions expressed canonical fibroblast markers such as THY1, PDPN, and FAP whilst immune-specific genes such as PTPRC, CD3E and MS4A1 were present within Immune regions (*Figure 2D and E*, *Figure 2—figure supplement 4*).

To align the regional transcriptional landscape to specific cell subsets, RNA expression profiles defined by NanoString DSP analysis were mapped onto an additional scRNA-Seq dataset derived from three additional patients (*Figure 2—figure supplement 5*; *Pearce et al., 2023*). This revealed that, whilst the great majority of stromal-associated genes were expressed from fibroblasts, the expression of *CSF1R* within stroma was largely derived from myeloid cells, *CTNNB1* localized to endothelial cells and expression of *KRT* was identified as *KRT18* within epithelial cells (*Figure 2—figure supplement 5C*).

## The transcriptional profile of stromal regions is strongly determined by proximity to tumor

We next went on to assess gene expression within stroma in relation to proximity to tumor (*Figure 3*). Transcriptional profiles were seen to vary markedly between tumor-proximal or tumor-distal ROI. In particular, expression of *DKK3* and *PDPN* was markedly increased in stroma-proximal regions (*Figure 3A, B and C*) and both are established markers of cancer-associated fibroblasts implicated in support of tumor growth (*Zhou et al., 2018*; *Hirayama et al., 2018*; *Shindo et al., 2013*). In contrast, expression of *C3, SFRP2, STAT3, IL-6* and *THY1* was increased in tumor-distal stroma. *C3* and *SFRP2* expression were particularly elevated (*Figure 3C*) and localization of C3 expression to fibroblasts was further suggested by correlation with fibroblast, but not monocyte, marker genes (*Figure 3—figure supplement 1*). This is noteworthy given the emerging importance for intracellular complement expression and the action of SFRP2 as a Wnt inhibitor. STAT3 and IL-6 expression could be explained by their presence within inflammatory CAFs whilst THY1 is commonly expressed on stem-like populations of fibroblasts (*Shi et al., 2019*). The stem cell marker CD34 was also expressed in this region (*Figure 3A, B and C*).

Immunohistochemical staining confirmed extreme polarization of podoplanin, DKK3 and C3 expression in relation to tumor proximity. Podoplanin was expressed on stroma that encased tumor whilst DKK3 expression was present both within tumor and tumor-proximal stroma. In contrast, expression of C3 was localized to distal stroma regions (*Figure 3D*).

## Mapping of spatial transcriptional profiles on to scRNA-Seq reveals key biochemical pathways associated with proximal and distant fibroblasts

Given the profound influence of tumor apposition on the NanoString profile of fibroblasts we were interested to explore global fibroblast transcriptome in relation to spatial localization. The minimal NanoString gene set defining proximal and distant fibroblast subsets was therefore explored within the scRNA-Seq dataset from three additional donors (*Figure 4A*). Two fibroblast clusters were observed from scRNA-Seq unsupervised clustering analysis and defined as sc-proximal and sc-distal

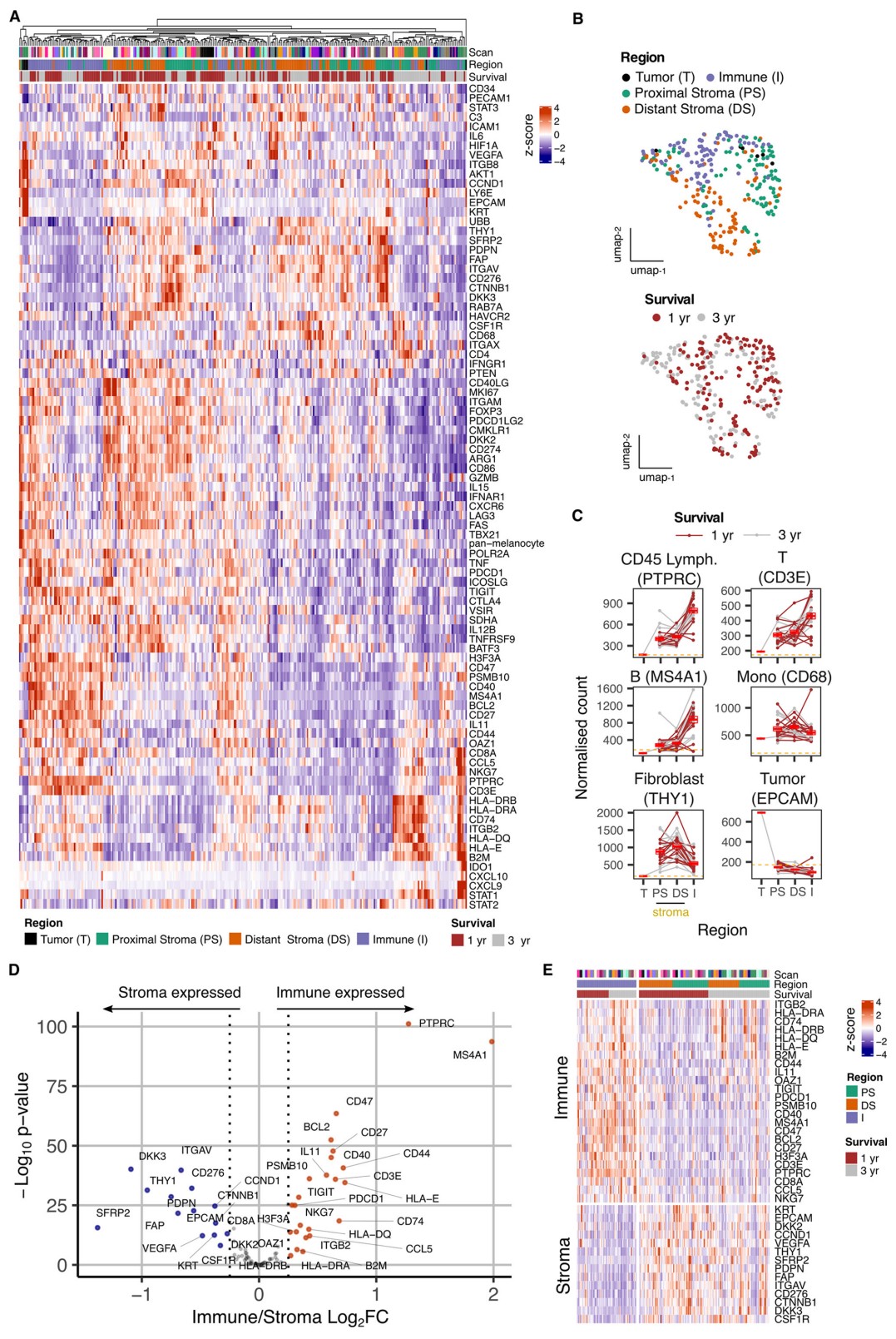

**Figure 2.** Overview of gene expression data from spatially-defined stromal and immune regions within the PDAC tumor microenvironment using NanoString GeoMx DSP. (**A**) Expression profile of all endogenous probes across regions of interest (ROI) with hierarchical clustering of ROIs. (**B**) UMAP embedding from normalized count data showing all ROIs overlaid with ROI-specific annotations of Region (Immune/Stroma/Tumor type) and Survival (1 yr/3 yr). (**C**) Mean normalized count of cell type marker genes within regions. Lines indicate regions from the same patient; dashed line represents

*Figure 2 continued on next page*

*Figure 2 continued*

mean background threshold from negative probes; Mean +/-SE of mean shown in red. (**D**) Differential expression analysis to identift genes expressed differentially between Immune and Stroma ROIs. Colored points indicate differentially expressed genes (DEG) (BH adjusted p<0.05 and absolute log$_2$FC >0.25). (**E**) Immune and Stroma expression signatures from DEGs identified in D.

The online version of this article includes the following figure supplement(s) for figure 2:

**Figure supplement 1.** Raw count expression profiles.

**Figure supplement 2.** Housekeeping gene correlations and data normalisation.

**Figure supplement 3.** Correlation matrix of endogenous probes.

**Figure supplement 4.** Individual gene expression profiles on UMAP embeddings of all regions of interest (ROIs).

**Figure supplement 5.** High level cell type contexture of PDAC tumor microenvironment.

populations due to their distinct proximal and distant gene expression signatures (*Figure 4B and C*). Transcriptional profiles were highly divergent between proximal compared to distal clusters with 47 genes differentially upregulated in the distal cluster and 36 genes differentially upregulated in the proximal cluster (*Figure 4D*). The sc-proximal clusters showed high expression of myofibroblast (myCAF) marker genes including *MMP11* and *HOPX* whilst the distal population was enriched for expression of genes associated with inflammatory CAF (iCAF) such as *CXCL12* and *CFD* (*Figure 4E*).

To investigate the likely functions and master transcription factor regulators that are active for each cluster, differentially enriched pathways, GO-terms and TF target gene sets were identified (*Figure 4F*). This showed that sc-proximal fibroblasts are enriched for cell division, chemotaxis and heat shock protein binding. Furthermore, they express a wide range of Wnt ligands including WNT5A, WNT11, WNT2, WNT5, WNT5A and WNT5B (*Figure 4F and G*). In contrast, sc-distal fibroblasts show enrichment of pathways associated with generation of the extracellular matrix, negative regulation of stem cell proliferation, complement activation and retinoic acid metabolism (*Figure 4F*). Also notable was expression of many members of the complement pathway as well as many genes associated with retinoic acid metabolism (*Figure 4G*). The relative gene set enrichment profiles highlight a potential further subcluster within the sc-distant cells which may indicate the presence of transitional cells at the interface between distant and proximal fibroblasts (*Figure 4F*).

Enrichments of transcription factor target gene sets (regulons) showed considerable divergence and reveals how spatially determined activity of transcription factors could underpin differential fibroblast programming.

## Podoplanin and hypoxia predict poor outcome whilst high level expression of immune regulatory genes associates with superior clinical outcome

The study cohort had been selected to comprise patients with poor or good clinical outcome to allow potential identification of spatial transcriptional correlates of disease progression. Poor outcome was defined as death within 1 year whilst patients with good outcome exhibited survival beyond 3 years (*Figure 5*).

Overall gene expression profiles were initially compared between these two groups to define spatially unaware prognostic transcriptional signatures. High level transcriptional expression of *PDPN*, *HIF1A*, *PDL1* (CD274), and *VEGFA* were associated with poor clinical outcome (*Figure 5A*, *Figure 5— figure supplement 1*). Ten genes were upregulated in patients with survival beyond 3 years and were characterized predominantly by immune activation with increased expression of MHC class I and class II, complement C3 and chemokines CCL5 and CXCL9. The integrin ITGB2 (CD18) and STAT1 also showed increased expression in this group.

The spatial expression of these prognosis-associated genes was then assessed across the two risk groups (*Figure 5B*). To further pinpoint the likely spatially-defined signal contribution of prognostic genes, good vs bad differentially expressed genes were identified from a within Immune region and within Stroma region analysis (*Figure 5C*). Additionally, within proximal and within distant stroma region analysis assessed the relative contribution to prognostic signals by proximity to tumor (*Figure 5D*).

Patients with a poor outcome expressed PDPN broadly across the tumor microenvironment whilst HIF-1α and VEGF expression also extended into the distal stromal and immune regions and likely indicates more extensive hypoxia in this subgroup. Proximal-to-distant stroma expression gradients

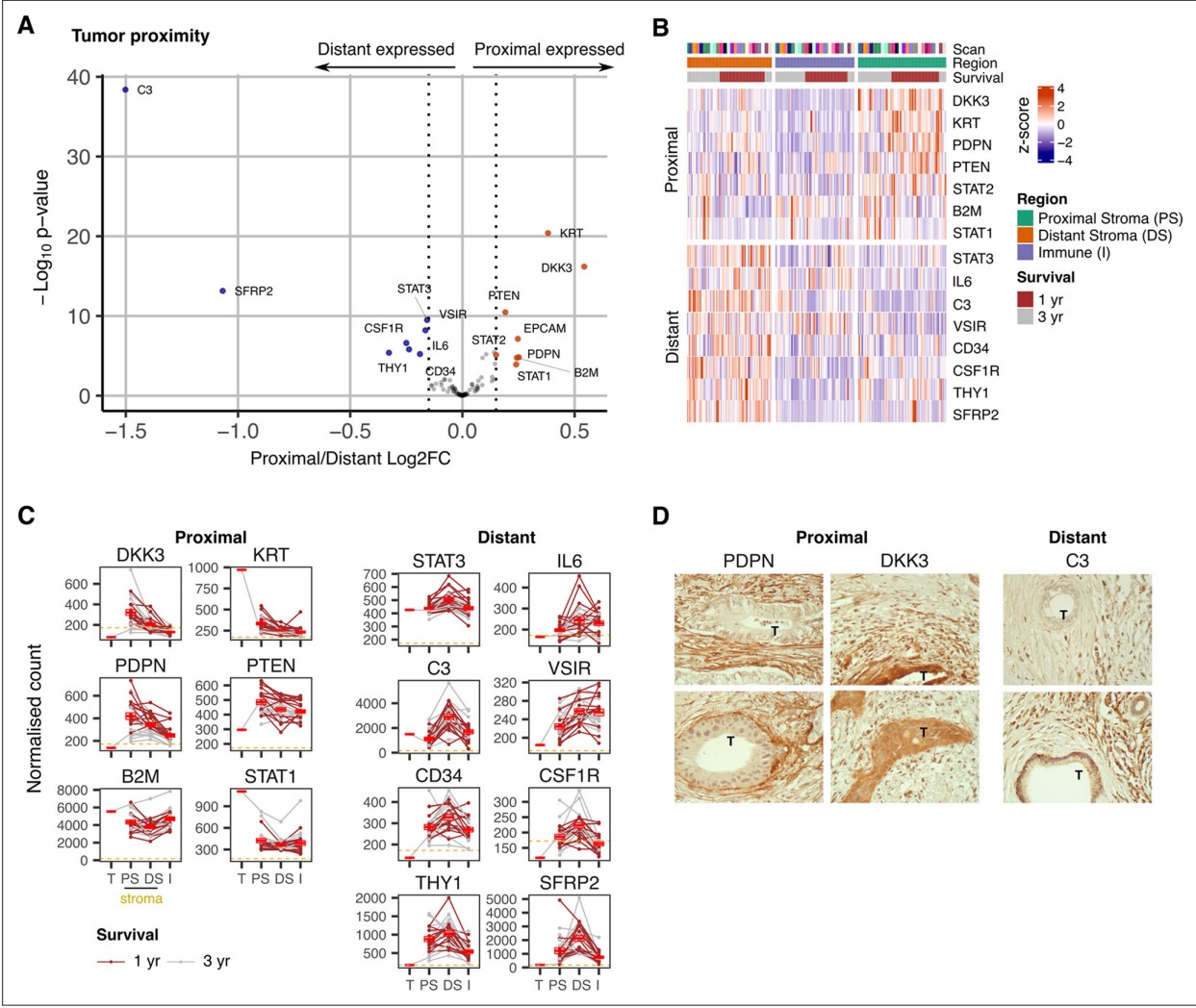

**Figure 3.** Expression signature of PDAC tumor-proximal and tumor-distal stromal cells. (**A**) Differentially expressed genes (DEGs) between stroma regions proximal (P) or distal (D) from tumor. Colored points indicate differentially expressed genes (BH adjusted $P<0.05$ & absolute $log_2FC >0.25$). (**B**) Stroma proximity-to-tumor expression signature from DEGs identified in A. (**C**) Relative expression of genes within four PDAC regions: Tumor (T), proximal-tumor stroma (PS), distal-tumor stromal (DS) and immune (I). Lines indicate paired regions from the same patient; dashed line represents mean background threshold from negative probes; Mean +/-SE of mean shown in red. Shown as within patient mean normalized count vs region type for DEG identified in A. (**D**) Representative immunohistochemical staining of podoplanin, DKK3 and C3 proteins in relation to tumor cells (T) in PDAC tissue.

The online version of this article includes the following figure supplement(s) for figure 3:

**Figure supplement 1.** Correlations of proximity specific markers DKK3 and C3.

could also be an indicator of prognosis with CD44, CCL5, EPCAM, and HIF1A all identified as having divergent gradients in Good vs Bad prognosis groups (*Figure 5—figure supplement 1C*).This further highlights the spreading of HIF1A as a feature of poor prognosis. Good prognosis was associated with broad expression of most immunostimulatory and immunoregulatory genes whilst expression of IL-11 and HLA-E was focused within distal stroma (*Figure 5C, D and E*). Expression of complement C3 and NKG7, a regulator of cytotoxic granule release (*Ng et al., 2020*) within the immune region was also enhanced in patients with good clinical outcome.

A graphical summary (*Figure 6*) of the combined Nanostring nCounter and scRNA data analysis highlights these key data-defined characteristics of spatially determined regions within the PDAC tumor microenvironment.

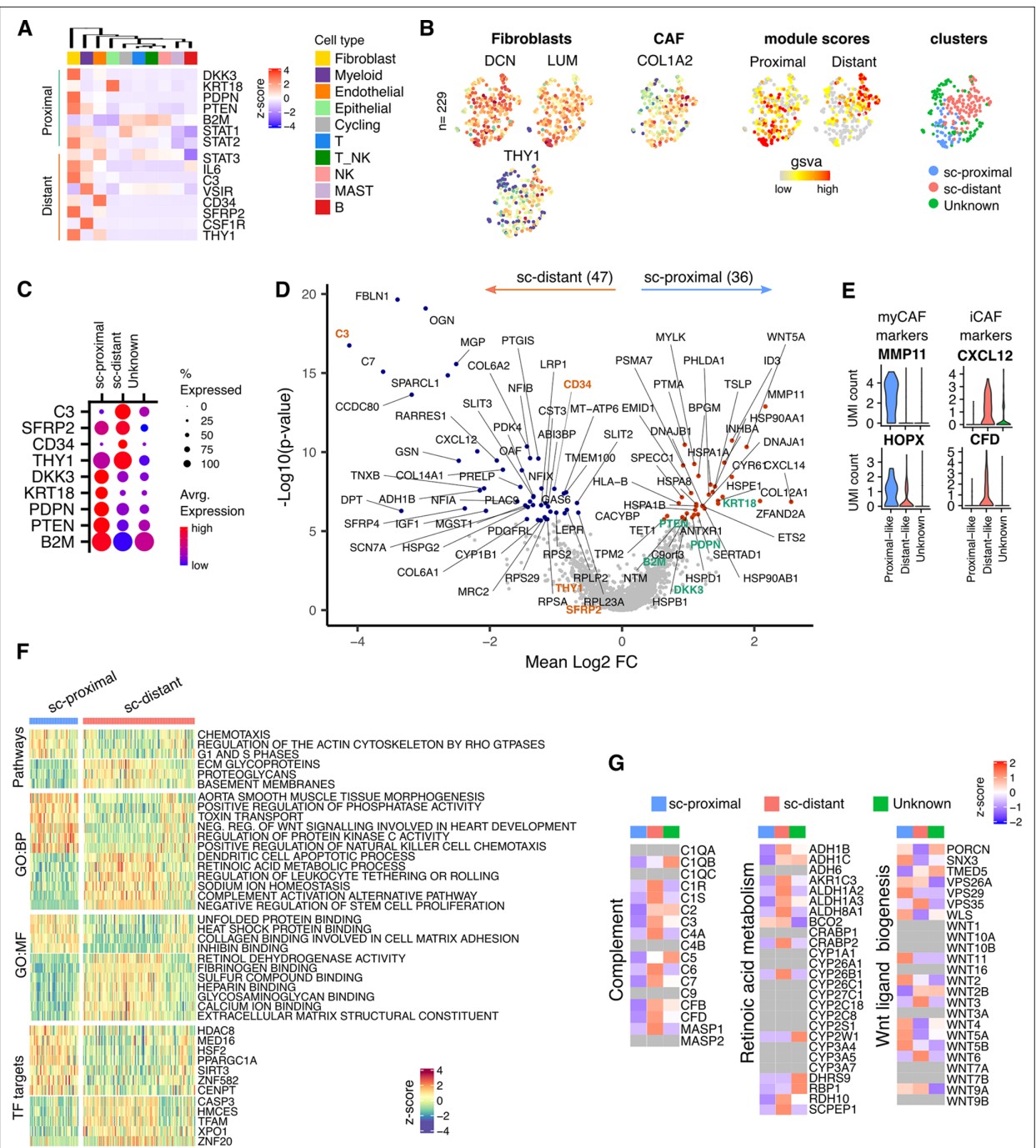

**Figure 4.** Proximal and Distal Fibroblast populations identified in single cell transcriptome data of PDAC. (**A**) Average expression of spatially defined tumor-proximal or tumor-distant stromal genes within cell types defined by scRNA-seq (n=3). (**B**) UMAP embedding of scRNA-Seq data from fibroblasts overlaid with (left to right) expression of the canonical fibroblast marker genes *DCN, LUM and THY1; COL1A2* found in CAFs; gene set variation analysis (GSVA) signature score for tumor-proximal (*DKK3, PDPN, PTENSTAT2, B2* and *STAT1*) or tumor-distal (*STAT3, IL6, C3, VSIR, CD34, CSF1R, THY1, SFRP2*) associated stromal genes; Clustering based on unsupervised Louvain assignment. n=229 Fibroblast cells.(**C**) Average cluster-wise expression profile of selected proximal and distant stroma associated genes as identified by spatial profiling. (**D**) Differential expression analysis between sc-proximal and sc-distant fibroblast cells. Colored points indicate differentially expressed genes (BH adjusted p<0.05 & absolute log₂FC >0.5). (**E**) Violin plots depicting cluster-wise expression distribution of canonical myCAF and iCAF marker genes. (**F**) GSVA score profiles identified as differentially enriched (BH adjusted p<0.001) in sc-distant vs sc-proximal cells. (**G**) Average within-cluster expression profile of Complement, Retinoic acid metabolism and Wnt ligand biogenesis gene sets. Grey = no detectable expression.

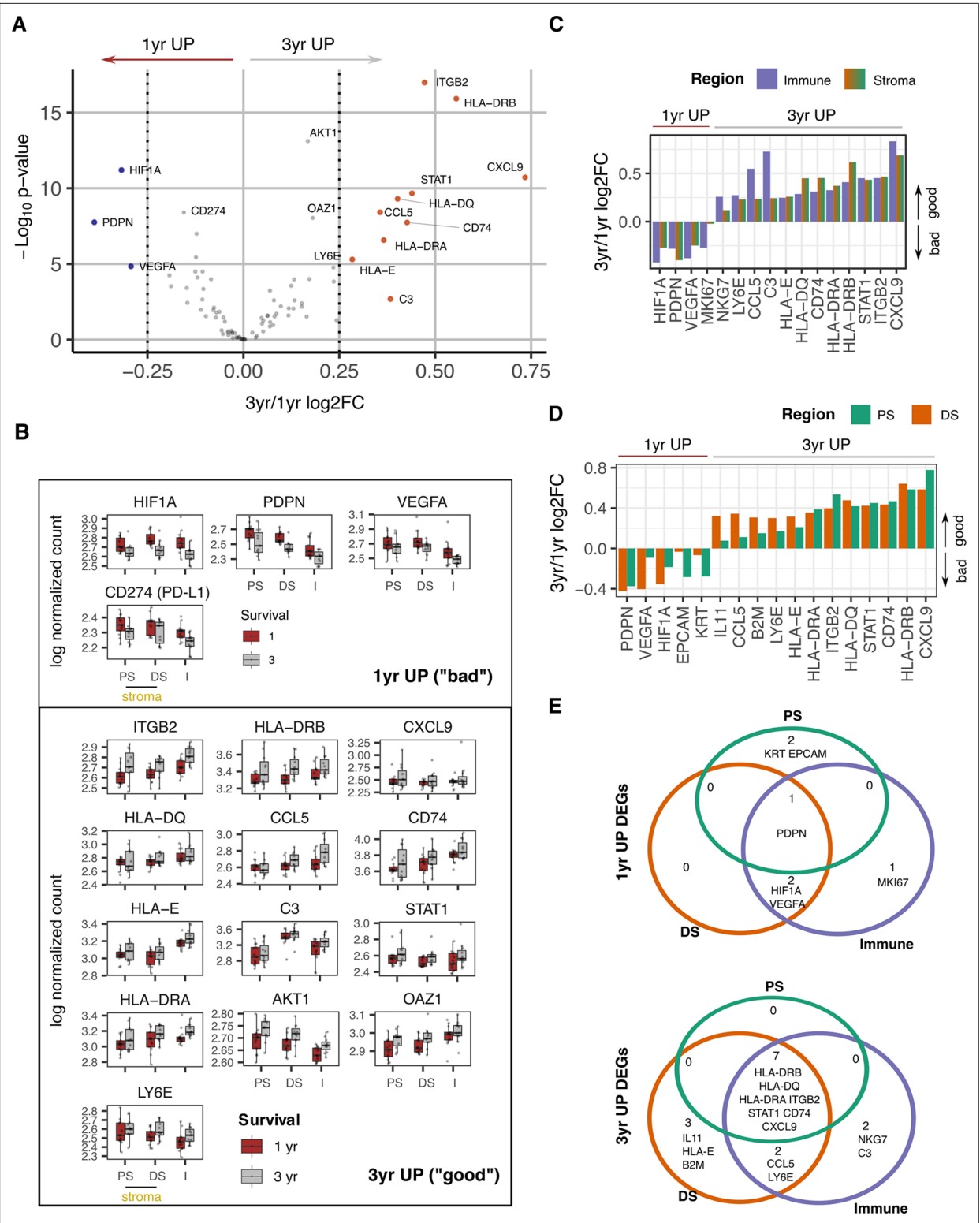

**Figure 5.** Survival expression signatures within spatially defined regions of PDAC. (**A**) Differential gene expression from all regions in relation to poor (<1 year) or good (3+year) survival. Coloured points indicate differentially expressed genes (BH adjusted p<0.05 and absolute log₂FC >0.25). (**B**) Regional expression of survival-associated genes identified in A. Mean +/-SE of mean. PS = Proximal Stroma; DS = Distant Stroma; I=Immune. (**C**) 3 yr/1 yr fold change in expression of survival-associated genes within Immune and Stroma regions. (BH adjusted p<0.05 and absolute log₂FC >0.25). (**D**) 3 yr/1 yr fold change in expression of survival-associated genes within Tumor-Proximal and Tumor-Distal regions. (BH adjusted p<0.05 and absolute

*Figure 5 continued on next page*

*Figure 5 continued*

log$_2$FC >0.25). (**E**) Venn displaying overlaps of 3 yr vs 1 yr survival DEGs (BH adjusted p<0.05 and absolute log$_2$FC >0.25) within tumor-proximal stroma (PS), tumor-distal stroma (DS) and Immune (I) regions.

The online version of this article includes the following figure supplement(s) for figure 5:

**Figure supplement 1.** Profile of Survival expression signatures within PDAC TME.

## Discussion

Increased understanding of the pancreatic ductal adenocarcinoma microenvironment is essential for the development of targeted therapies. Here, we combined spatial and single cell transcriptomic analysis to interrogate patterns of cellular transcription in relation to tumor proximity and related this to clinical outcome. This reveals spatially determined transcriptional programming of fibroblasts with potential opportunities for therapeutic development.

Proximity to tumor was seen to be a strong determinant of transcriptional activity of stromal cells. In particular, DKK3 and PDPN were both increased markedly on tumor-proximal cells. PDPN expression is strongly enhanced on cancer-associated fibroblasts in PDAC (*Shindo et al., 2013*) and high expression levels are correlated with poor prognosis in some, but not all, studies (*Mezawa and Orimo, 2016*). DKK3 is a Wnt regulator and is emerging as a potentially important therapeutic target (*Zhou et al., 2018*). A range of genes showed increased expression within stromal populations distal from

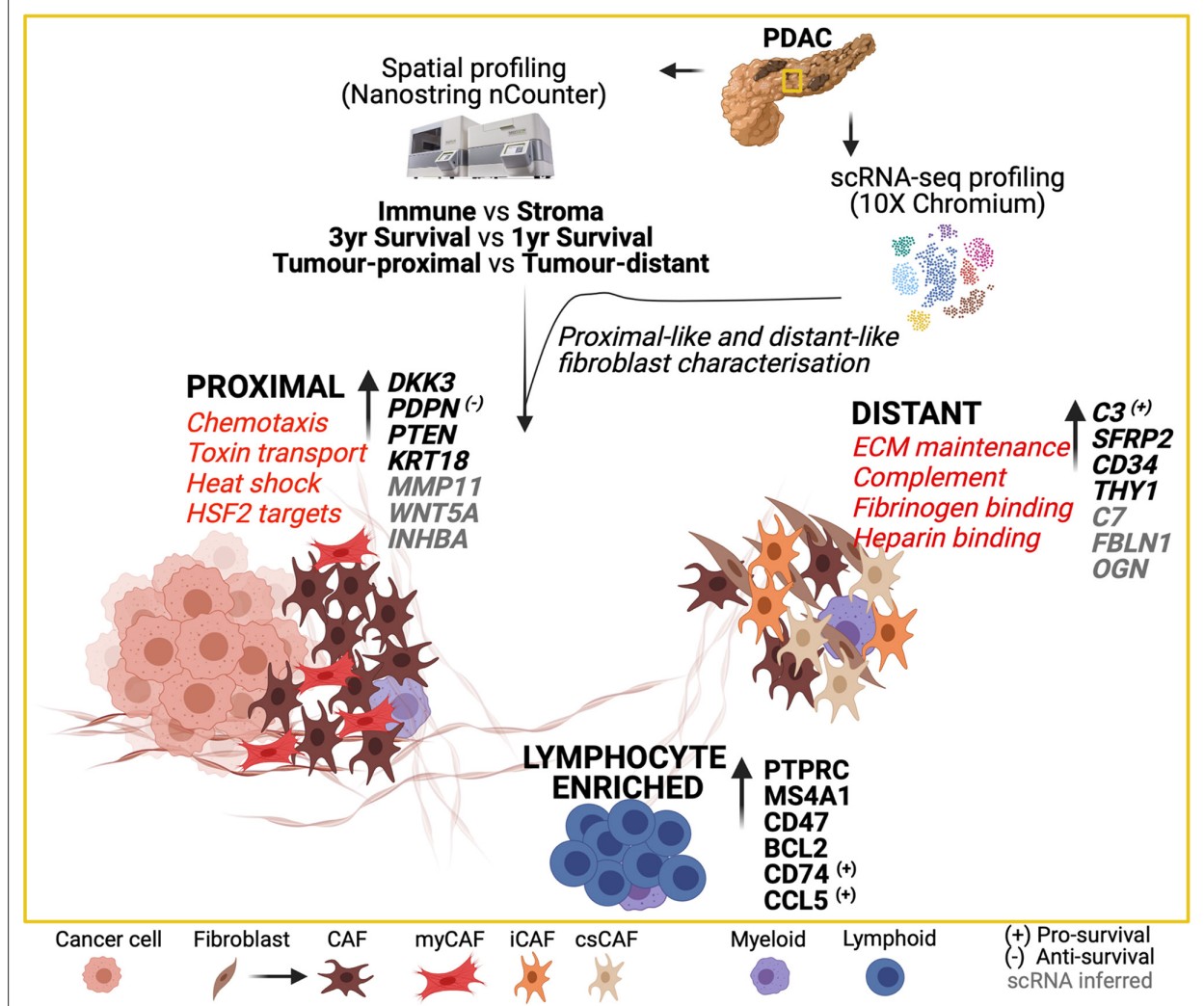

**Figure 6.** Graphical summary of the transcriptional features of spatially defined regions in the PDAC tumor microenvironment.

tumor including the C3 component of complement and SFRP2, a soluble modulator of Wnt signaling. CD34, a marker of stromal stem cells, was also expressed more highly in this region and may indicate spatial differentiation of stromal cells towards the tumor. Due to the nature of two-dimensional imaging, we cannot rule out that cancer cells may be present above or below the plane of the tissue section. Nonetheless, clear differences between stromal populations were observed suggesting that this potential occurrence did not significantly impact our analysis.

Integration of spatially defined transcription signatures with an additional single cell RNA-Seq dataset allowed development of a transcriptional atlas of proximal and distal stromal cells. Tumor-proximal populations displayed features typical of myofibroblasts whilst more distal populations had an inflammatory profile, in line with previous reports (*Öhlund et al., 2017*). Myofibroblast markers included the potent pro-tumorigenic chemokine CXCL14 (*Augsten et al., 2014*) and WNT5A which may contribute to the differentiation of adipocytes to CAFs (*Zoico et al., 2016*). Indeed, Wnt ligand signaling plays a key role in PDAC progression and therapeutic resistance, and tumor-proximal fibroblasts are seen to be strong contributors to the Wnt ligand pool with high level expression of the ligands WNT5A, WNT11, WNT2, WNT5, WNT5A, and WNT5B. Transcriptional regulation of cell division was also increased, suggesting enhanced proliferation of stromal cells when locally exposed to tumor and in line with prior reports (*Kalluri and Zeisberg, 2006*).

In contrast, stromal cells located more distally from tumor retained functions such as generation of extracellular matrix proteins. Striking expression of a wide range of complement proteins was also seen at this site. CAF populations expressing complement proteins have been observed previously in PDAC (*Chen et al., 2021a*) and overlap with the transcriptional profile of inflammatory CAF. However, there remains debate as to their potential additional expression within myeloid lineages and high dimensional immunofluorescence analysis may help to resolve this. The physiological role of intracellular complement expression is receiving considerable interest with evidence that it may impact on immune surveillance in pre-clinical models (*Kwak et al., 2018*). THY1 is not a canonical fibroblast marker but is typically expressed on subsets of myofibroblasts and here we also observed increased levels in the tumor-distal region. Interestingly, we find expression of inflammatory genes in some tumor distal α-SMA+ regions and it is tempting to speculate that this could represent a potential hybrid myofibroblastic/inflammatory CAF state.

A further finding of note was increased expression of a range of genes associated with retinoids. Vitamin A-containing lipid droplets are known to be enriched within quiescent pancreatic stellate cells in close proximity to the basal aspect of pancreatic acinar cells (*Erkan et al., 2012*) and, as such, the increased retinoic acid signature in tumor-distal fibroblasts could indicate that these are less activated. Indeed, patients with PDAC are often vitamin A deficient whilst retinoic acid treatment can suppress stellate cell proliferation with associated reduction in Wnt-β-catenin signaling and localized tumor apoptosis (*Froeling et al., 2011*). ATRA treatment has been shown to be tolerable in patients with advanced disease and is under investigation in phase I trials (*Kocher et al., 2020*). Distal populations were also enriched for expression of genes associated with negative regulation of stem cell proliferation and may indicate a potential role for cells within this environment in limiting tumor cell progression.

CAF populations exhibit extreme plasticity and factors such as IL-1 and TGF-β are emerging as important mediators of local phenotype (*Biffi et al., 2019*). Analysis of relative transcription factor binding expression within tumor-proximal or distal stroma identified substantial differences in transcription factor activity at the two sites. A wide range of transcription factor targets were differentially expressed and indicate the importance of local cellular environments in exploiting the transcriptional plasticity of fibroblasts.

The study cohort had been selected to include patients with poor or good clinical outcome, based on survival below 1 year or above 3 years, respectively. High level expression of podoplanin, HIF-1α and VEGF were associated with poor outcome. The negative prognostic impact of podoplanin expression in PDAC has been documented previously (*Mezawa and Orimo, 2016*) and podoplanin-positive stromal cells enhance invasion and proliferation of tumor cells. However, downregulation of podoplanin expression does not reverse this effect indicating an important role for additional pathways within this population (*Shindo et al., 2013*). PDAC disease progression following surgical resection is usually related to metastasis or local progression and the potential role of different fibroblast subsets in the development of the metastatic niche requires further investigation (*Xu et al., 2010*).

Expression of HIF-1α is reflective of the hypoxic environment within PDAC tumors and indicates that the intensity of hypoxia is an independent determinant of clinical outcome (*Ye et al., 2014*; *Chen et al., 2021b*; *Hao, 2015*). Indeed, spatial extension of HIF-1α expression into distal stroma and immune microenvironments was an additional risk factor and indicates that the breadth of hypoxia is of prognostic importance. HIF-1α expression in PDAC is associated with a range of features including enrichment of glycolysis, modulation of mTORC1 and MYC signaling, and immune suppression (*Zhuang et al., 2021*; *Zhao et al., 2014*). As such, this represents a challenging tumor subgroup for therapeutic intervention, although the introduction of HIF-1α inhibitors offers encouragement in this regard (*Semenza, 2023*). Hypoxia is also likely to explain increased levels of VEGF expression in patients with poor prognosis. VEGF-targeted therapies have not shown significant utility in PDAC but could potentially be considered as part of a multi-modal therapeutic approach (*Cabebe and Fisher, 2007*).

In contrast, many immunoregulatory and immunostimulatory genes were increased in patients with good prognosis and concur with studies showing that the extent of lymphocytic infiltration is a favorable indicator for outcome. Liudahl et al. used chromogen-based multiplexed immunohistochemistry (mIHC) to generate an atlas of leucocyte contexture within PDAC (*Liudahl et al., 2021*) and extended prognostic utility to immune subpopulations. It was noteworthy that elevated expression of HLA class II genes was seen in patients with longer term survival and as this association extended into stromal regions it may indicate an important role for HLA-DR +antigen-presenting CAF (ApCAF) populations (*Elyada et al., 2019*). Expression of complement protein C3 was associated with good clinical outcome and indicates that this pathway can also help to contain tumor growth (*Revel et al., 2020*) despite early indications of a potential pro-tumorigenic role (*Kwak et al., 2018*). Indeed, the beneficial effect of ApCAF in lung cancer is mediated partially through expression of complement proteins which rescue intratumoral T cells from exhaustion (*Kerdidani et al., 2022*) and this may provide a unifying explanation for the prognostic value of HLA class II and complement expression in this study.

Expression of IL-11 within distal stroma was also a positive prognostic sign and is noteworthy given a previous report of a similar association with elevated serum concentrations (*Ren et al., 2014*). IL-11 is an inflammatory protein within the IL-6 family and as such further analysis of the mechanisms by which it can help to contain PDAC development would be valuable. High level expression of NKG7 within immune regions was also beneficial and, given its central role in regulation of cytotoxic granule release (*Ng et al., 2020*), this is noteworthy given its emerging role as a predictive factor in response to checkpoint protein inhibition (*Wen et al., 2022*). Overall, the transcriptional correlates of good prognosis clearly identify immunological processes as the central determinant of clinical outcome and augur well for therapeutic interventions that can unmask this immune potential. Immune checkpoint inhibition has been largely unsuccessful for this patient subgroup but there is clearly latent immunogenicity within the PDAC microenvironment and the use of agonistic anti-CD40 antibodies has shown promise in clinical studies (*Byrne et al., 2021*).

A limitation of the study is that tumors are markedly heterogeneous and as such our findings may not be representative of the complete architecture. However, to overcome this we used a broad patient cohort and selected regions of interest from across the biopsies. Multiplex immunohistochemical analysis will also be of value to confirm co-expression of proteins within cell subsets.

In conclusion, we find that transcriptional activity of stromal subsets is strongly regulated by their relative proximity to tumor and define the transcriptional landscape in relation to spatial localization. Hypoxia is a correlate of poor outcome whilst approaches to enhance the inflammatory environment of distal stroma could offer strategies to improve the clinical outcome for this patient group. Indeed, successful therapy for PDAC may require multi-modal approaches such as HIF inhibition with immune checkpoint blockade (*Salman et al., 2022*) or personalized vaccine regimens (*Rojas et al., 2023*).

# Materials and methods

## Participants

FFPE tissue from 25 treatment-naïve patients undergoing pylorus-preserving pancreaticoduodenectomy (PPPD) who presented with localized disease were selected for this study. Samples were obtained from the Birmingham Human Biomaterials Resource Centre HBRC (HTA Licence 12358)

**Table 1.** Morphology marker antibodies.

| Name | Channel | Host | Company | Clone # | Catalog # | Concentration used |
|------|---------|------|---------|---------|-----------|--------------------|
| SMA | 488 | Mouse | Invitrogen | 1A4 | 53-9760-82 | 1:200 |
| Syto83 | 532 | | Thermo fisher | | | 400 nM |
| PanCk | 594 | Mouse | Novus | AE1/AE3 | NBP2-33200DL594 | 1:500 |
| CD45 | 647 | Mouse | Novus | 2B11+PD7/26 | NBP2-34528AF647 | 1:200 |

ethically approved North West - Haydock Research Ethics Committee; Ref 20/NW/0001, local ethics number 18–304.

## Sample processing

FFPE tissue blocks were sectioned at 5 µm thickness, deparaffinized and rehydrated using conventional methods. The slides were profiled using NanoString GeoMx Digital Spatial RNA Profiling (DSP) platform through the Technology Access Program (TAP) by NanoString (Seattle, WA, USA). Briefly, immunofluorescent antibody staining was performed with tissue morphology markers α-SMA, Syto83, Pan-CK ,and CD45 (*Table 1*).

In parallel, slides were stained with a panel of photocleavable RNA probes. Custom regions of interest (ROI) were selected based on these markers to generate specific domains including 'tumor-proximal stroma', tumor-distal stroma' and 'immune enriched' areas. 'Tumor-proximal stroma' refers to regions within the tumor that are surrounded by tumor epithelium, while "tumor-distal stroma" refers to regions that are located as far away as possible from malignant ducts and lack surrounding epithelium.

To minimise the confounding effect of tumor heterogeneity, four ROI were selected for each domain per slide. UV-cleavable probes within each ROI were liberated by UV light, hybridized to optical fluorescent barcodes then counted on the nCounter to determine the absolute number of mRNA transcripts.

## NanoString nCounter data analysis

Raw NanoString nCounter data expression matrix (*Source data 1*) was processed following the normalization and quality control procedures as described elsewhere (*Bhattacharya et al., 2021*). Due to a redundancy in the tags used for both IFNG and ACTA2, data from these probes had to be removed from further analysis. Correlations of housekeeping gene expression across all ROIs were assessed to select the most correlated housekeeping probes H3F3A and UBB to use for downstream normalization (Supp. Figure S2A). Unwanted variation was removed using the R package RUVSeq (*Risso et al., 2014*). Firstly, distributional differences were scaled between lanes using upper-quartile normalization then unwanted technical factors were estimated in the resulting gene expression data with the RUVg function selecting H3F3A and UBB as the negative control genes and the number of dimensions of unwanted variation to remove set to 1. A variance stabilizing transformation of the original count data was computed using DESeq2 (*Love et al., 2014*) and estimated unwanted variation was removed using the removeBatchEffects function from limma (*Ritchie et al., 2015*). RLE plots were used to detect any potential outliers before and after normalization (Supp. Figure S2B).

Differential expression analysis was conducted to compare Immune vs stromal regions, 3 yr vs 1 yr survival and tumor-proximal vs tumor-distant stromal regions using DESeq2, adjusting for multiple testing with Benjamini-Hochberg (BH) procedure. Differentially expressed genes were determined by BH adjusted $p<0.05$ and absolute log2FC $>0.25$.

Dimensionality reduction by Uniform Manifold Approximation and Projection (UMAP) was performed on the normalized counts matrix with the umap R package and ggplot2 utilized for plotting. Heatmap visualizations were generated using the ComplexHeatmap package. Pearson correlation was calculated and plots generated using ggpairs and ggcorr functions from the R package GGally.

### scRNA-Seq data analysis

Genes of interest identified from nCounter data analysis were further explored for their expression profiles in single-cell RNA sequencing data (GEO accession GSE210199) of cells within the tumor microenvironment of 3 PDAC patients (*Pearce et al., 2023*).

### Raw read data processing

Raw reads were processed using CellRanger (10X Genomics, v3) functions mkfastq and count. Raw bcl files were converted to fastq and aligned to the human reference genome GRCh38. Gene expression matrices for each patient were analyzed by R software (v3.6). Data pre-processing, QC, dimensionality reduction, clustering and subsequent downstream analysis was performed using the Seurat package (v3.1.1).

### Data integration and clustering

Data from 3 PDAC patient samples was integrated following Seurat SCTransform Integrate Data workflow using the top 3000 most variable genes as integration features. Principal Component Analysis (PCA) was applied and Uniform Manifold Approximation and Projection (UMAP) embedding determined using PCs 1:20. For unsupervised clustering, a shared nearest neighbour graph based on Euclidean distance in PCA space was constructed using Seurat FindNeighbours function and the modules within this graph representing clusters were identified using the Louvain algorithm with Seurat FindClusters.

To annotate clusters with high-level cell type, canonical cell type marker gene expression level was assessed.

### scRNA-Seq Fibroblast data analysis

Transcriptome data was subset taking Fibroblast cells only and unsupervised clustering re-applied on Fibroblasts alone. Expression profile of stromal expressed genes identified from the nCounter dataset to be associated with tumor proximal or tumor-distant regions was assessed within the Fibroblast scRNA-Seq data. These tumor-proximal and tumor-distant gene signatures were scored using GSVA to assess likely tumor-proximal and tumor-distant fibroblasts. Expression profiling and GSVA signature scoring were used to annotate fibroblast subpopulations identified through clustering as 'Proximal-like' and 'Distant-like'.

To expand the pool of possible transcriptional markers for tumor proximal and tumor distant fibroblasts, differential expression analysis was conducted comparing Proximal-like and Distant-like clusters using findMarkers with MAST option (test.use = 'MAST'), which uses a hurdle model tailored to scRNA-Seq data. MAST is a two-part GLM that simultaneously models how many cells express the gene by logistic regression and the expression level by Gaussian distribution (*Finak et al., 2015*). Differential expression testing was performed using the likelihood ratio test. Differentially expressed genes were determined by Benjamini Hochberg adjusted $p<0.05$ and absolute log2FC $>0.5$.

## Acknowledgements

This research was supported by a Cancer Research UK (CRUK) Programme Grant CRUK-A21135.

# Additional information

## Funding

| Funder | Grant reference number | Author |
|---|---|---|
| Cancer Research UK | A21135 | Wayne Croft<br>Hayden Pearce<br>Sandra Margielewska-Davies<br>Lindsay Lim<br>Samantha M Nicol<br>Gary Middleton<br>Keith Roberts<br>Paul Moss |
| European Union Horizon 2020 Marie Skłodowska-Curie Actions PAVE | 861190 | Fouzia Zayou |

This independent research was carried out at the National Institute for Health and Care Research (NIHR) Birmingham Biomedical Research Centre (BRC). The views expressed are those of the author(s) and not necessarily those of the NIHR or the Department of Health and Social Care. The funders had no role in study design, data collection and interpretation, or the decision to submit the work for publication.

## Author contributions

Wayne Croft, Conceptualization, Formal analysis, Supervision, Investigation, Visualization, Methodology, Writing - original draft, Writing - review and editing; Hayden Pearce, Conceptualization, Formal analysis, Supervision, Investigation, Methodology, Writing - original draft, Writing - review and editing; Sandra Margielewska-Davies, Lindsay Lim, Investigation, Methodology, Writing - review and editing; Samantha M Nicol, Reena Merard, Rachel M Brown, Methodology, Writing - review and editing; Fouzia Zayou, Francesca Marcon, Sarah Powell-Brett, Brinder Mahon, Writing - review and editing; Daniel Blakeway, Formal analysis, Writing - review and editing; Jianmin Zuo, Investigation, Writing - review and editing; Gary Middleton, Keith Roberts, Funding acquisition, Writing - review and editing; Paul Moss, Conceptualization, Supervision, Funding acquisition, Writing - original draft, Writing - review and editing

## Author ORCIDs

Wayne Croft http://orcid.org/0000-0001-6780-5944
Hayden Pearce http://orcid.org/0000-0002-8380-8122
Sandra Margielewska-Davies http://orcid.org/0000-0002-5115-470X
Lindsay Lim http://orcid.org/0000-0003-1394-3297
Daniel Blakeway http://orcid.org/0000-0001-9501-7451
Francesca Marcon http://orcid.org/0000-0001-7439-8291
Paul Moss http://orcid.org/0000-0002-6895-1967

## Ethics

Samples were obtained from the Birmingham Human Biomaterials Resource Centre HBRC (HTA Licence 12358) ethically approved North West - Haydock Research Ethics Committee; Ref 20/NW/0001, local ethics number 18-304.

## Decision letter and Author response

Decision letter https://doi.org/10.7554/eLife.86125.sa1
Author response https://doi.org/10.7554/eLife.86125.sa2

# Additional files

## Supplementary files

• Supplementary file 1. NanoString nCounter RNA hybridisation probeset. List of RNA probe panels used for Nanostring nCounter data collection including the immunoncology core panel, fibroblast-specific, housekeeping and negative control probe sets.

- MDAR checklist
- Source data 1. Raw NanoString nCounter data.

### Data availability
*Source data 1* contains the raw Nanostring nCounter data.

The following previously published dataset was used:

| Author(s) | Year | Dataset title | Dataset URL | Database and Identifier |
|---|---|---|---|---|
| Pearce H, Croft W, Nicol S, Margielewska-Davies S, Powell R, Cornall R, Davis SJ, Marcon F, Pugh M, Powell-Brett S, Brown R, Middleton G, Mahon B, Fennell E | 2023 | Tissue-resident memory T-cells in pancreatic ductal adenocarcinoma co-express PD-1 and TIGIT and inhibition is reversible by dual antibody blockade | https://www.ncbi.nlm.nih.gov/geo/query/acc.cgi?acc=GSE210199 | NCBI Gene Expression Omnibus, GSE210199 |

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
