## [Editor Report]

The plasticity and heterogeneity of fibroblasts in the tumor microenvironment of pancreatic ductal adenocarcinoma (PDAC) has emerged as a key factor in determining tumor growth and therapeutic response. Here the authors use innovative approaches to combine spatial profiling with single cell transcriptomics to define tumor-proximal populations of fibroblasts that predict clinical outcome. Specifically, elevated expression of HIF-1a and podoplanin predicted worse outcome while inflammatory gene expression correlated with increased survival, suggesting future interventions targeting proximal fibroblast populations to mitigate against PDAC.

---

## [Decision Letter]

**Decision letter after peer review:**

Thank you for submitting your article "Spatial determination and prognostic impact of the fibroblast transcriptome in pancreatic ductal adenocarcinoma" for consideration by *eLife*. Your article has been reviewed by 2 peer reviewers, and the evaluation has been overseen by a Reviewing Editor and Tony Ng as the Senior Editor. The following individual involved in the review of your submission has agreed to reveal their identity: Marina Pasca di Magliano (Reviewer #2).

Essential revisions:

1) It is essential to add the use of additional fibroblast and myeloid markers, as suggested by both reviewers.

2) If possible, please add an increased analysis of the transcriptomics to include more cells and markers.

3) Please address reviewers' concerns with interpretation, proper citation, and general text accuracy and presentation.

*Reviewer #1 (Recommendations for the authors):*

1) The writing could be improved across the manuscript. There are often words or letters missing in the text.

2) The abstract reads as if the authors generated both the spatial and single-cell transcriptomics data by themselves from the same patients. This is misleading and should be amended.

3) Interestingly, the authors find expression of inflammatory genes in tumor distal, aSMA+ regions. This could indicate the existence of a potential hybrid myofibroblastic/inflammatory CAF state.

4) It would have been helpful if the authors included other canonical fibroblast markers in their nCounter analysis, such as PDGFRA, aSMA, and COL1A1. THY1 is not necessarily a canonical fibroblast marker and is rather expressed in subsets of myofibroblasts.

5) Line 198/199 – THY1 is repeated.

6) Line 236 – positive correlation of C3 expression with THY1 expression is NOT a confirmation for it to be fibroblast-derived, but rather an indication. Also, only THY1 but not other fibroblast marker genes (as mentioned in line 237) was used in the correlation analysis.

7) Line 239 – STAT3 and IL6 expression COULD be explained by their presence within inflammatory CAFs (their presence on iCAFs does not explain it).

8) The retinoic acid signature observed in tumor-distal fibroblasts could also indicate that these cells are less activated, given that quiescent stellate cells express high levels of retinoic acid metabolism genes.

9) In Figure 4F it looks like there are two distinct clusters within the sc-distant cluster, and the second one looks much more like the sc-proximal cluster. The authors should explain or represent this better.

10) Sup. Figure 7C is never mentioned in the text and is confusing. The authors should incorporate it in the text and explain what is shown in this panel.

*Reviewer #2 (Recommendations for the authors):*

Specific comments:

1) The authors select regions of interest (ROIs) that are proximal or distant from cancer cells; a more specific definition of how "proximal" and "distal" are defined would be useful. In addition, there are two potential issues to address or at least discuss: 1) the 2D nature of a tissue section makes it impossible to determine whether cancer cells are present in the same area but above or below the plane of the section. 2) the samples are described as biopsies of resected tumors: the authors should discuss whether the biopsy is likely to reflect the whole tumor or discuss the potential confounding effect of tumor heterogeneity.

2) Some of the markers of the distal stroma are also expressed by other cell types – for instance, complement genes are associated with macrophages. In order to confirm that they are indeed expressed in fibroblasts, the authors could consider co-immunofluorescent staining with lineage markers for fibroblasts and myeloid cells, respectively.

It is not clear what the take home of Figure 4F is; what kind of differences in transcription factor signatures distinguish the different fibroblast populations is not clear.

3) Figure 5 seems somewhat disjointed from the rest of the manuscript – it is unclear whether gene signatures that predict poor outcomes are linked to a specific geographic distribution of different types of fibroblasts, or whether it is simply the prevalence of each type that affects tumor characteristics.

4) The authors should discuss their findings in light of recent literature that either supports or contradicts the link between myCAFS and outcome, as this is an area of active investigation.

5) In Figure 2B, the colors of the tumor and proximal stroma are difficult to distinguish from one another.

6) Figure 2 (and other figures as well) many of the graphs and text is too small to clearly read. For the heat map, one option would be to highlight key genes labelling them with a larger font. The whole list could be moved to supplemental information. Some of the other graphs could be enlarged.

---

## [Author Response]

Essential revisions:1) It is essential to add the use of additional fibroblast and myeloid markers, as suggested by both reviewers.

Thank you for this advice. We have now added a range of new fibroblast and myeloid genes in the analysis using the extensive transcriptional data. This work is fully discussed in the revision and we are confident that this enhances the discovery outcomes considerably.

Unfortunately, further antibody or RNA probes cannot be added to the Nanostring experimental study as this is fully completed.

2) If possible, please add an increased analysis of the transcriptomics to include more cells and markers.

Thank you, we have now added additional fibroblast markers to Figure 4 confirming that the fibroblast populations are indeed expressing the canonical fibroblast marker genes *DCN*, *LUM* and *THY1. LUM* in particular has been shown to be highly specific to PDAC fibroblast cells (PMID: 34035226) and consistent with this, we observe high expression of this marker in our fibroblast scRNA data. Additionally, we have included the generic CAF marker gene *COL1A2* which again shows widespread expression in our fibroblast population.

3) Please address reviewers' concerns with interpretation, proper citation, and general text accuracy and presentation.

Thank you. We have substantially revised the manuscript and added a range of important additional citations.

Reviewer #1 (Recommendations for the authors):1) The writing could be improved across the manuscript. There are often words or letters missing in the text.

We thank the reviewer for this comment and have edited the manuscript to make the required corrections to these oversights.

2) The abstract reads as if the authors generated both the spatial and single-cell transcriptomics data by themselves from the same patients. This is misleading and should be amended.

Thank you for this point for which we apologise. This has now been amended.

3) Interestingly, the authors find expression of inflammatory genes in tumor distal, aSMA+ regions. This could indicate the existence of a potential hybrid myofibroblastic/inflammatory CAF state.

Thank you for this insight which may be reflective of the remarkable plasticity of fibroblasts and could represent an important transition subset. We have now added this information to the Discussion.

4) It would have been helpful if the authors included other canonical fibroblast markers in their nCounter analysis, such as PDGFRA, aSMA, and COL1A1. THY1 is not necessarily a canonical fibroblast marker and is rather expressed in subsets of myofibroblasts.

We thank the reviewer for highlighting the limitation of reliance on a single marker for a heterogenous population of cells. However, we would point out that we have used THY1, FAP and PDPN to identify fibroblast subsets.

We agree that THY1 is not a classical fibroblast marker and we have added this to the text. However, we find (Figure 4C) that THY1 is expressed in the majority of fibroblasts (as identified in the scRNAseq dataset) with average expression level that is lower in tumor-proximal stroma.

5) Line 198/199 – THY1 is repeated.

Thank you, this has now been corrected.

6) Line 236 – positive correlation of C3 expression with THY1 expression is NOT a confirmation for it to be fibroblast-derived, but rather an indication. Also, only THY1 but not other fibroblast marker genes (as mentioned in line 237) was used in the correlation analysis.

Thank you for highlighting this important point and we agree that this correlation alone is not confirmation. We have edited the text to read that the correlation of C3 with THY1, SFRP2, PDPN and FAP expression, and concurrent lack of correlation with CD68, is an indication that the C3 expression is likely to be fibroblast, rather than monocyte, derived. It is reassuring that significant positive correlation is specifically observed within distant stroma regions between C3 and additional Fibroblast marker genes (SFRP2 PDPN and FAP) and these have also been added to Figure 3 —figure supplement 1.

7) Line 239 – STAT3 and IL6 expression COULD be explained by their presence within inflammatory CAFs (their presence on iCAFs does not explain it).

Thank you, we have reworded this sentence to reflect that STAT3 and IL6 expression could be explained by presence within iCAFs.

8) The retinoic acid signature observed in tumor-distal fibroblasts could also indicate that these cells are less activated, given that quiescent stellate cells express high levels of retinoic acid metabolism genes.

We thank the reviewer for this interesting insight. We have now added this point to the Discussion.

9) In Figure 4F it looks like there are two distinct clusters within the sc-distant cluster, and the second one looks much more like the sc-proximal cluster. The authors should explain or represent this better.

We thank the reviewer for this comment. We agree that the GSVA profiles might suggest a potential subcluster within the sc-distant cluster that has a profile resembling the sc-proximal cells. The initial clusters were generated in an unsupervised manner to avoid any bias in defining the number of clusters that should be produced. One potential explanation is that this represents a transitional fibroblast state at the interface of the distal and proximal clusters. We have added an explanation of this observation to the text.

10) Sup. Figure 7C is never mentioned in the text and is confusing. The authors should incorporate it in the text and explain what is shown in this panel.

Thank you, and we apologise for the lack of reference to figure within the text.

This supplementary figure panel (now labelled as Figure 5 —figure supplement 1C) shows an analysis of proximal-to-distant expression gradients in the stromal regions as it is possible that spatial gradients in expression may also provide indication of prognosis. Indeed, we identify genes whose expression gradients are significantly different when stratified by 1 vs 3 year survival. This is now referred to and explained in the text.

Reviewer #2 (Recommendations for the authors):Specific comments:1) The authors select regions of interest (ROIs) that are proximal or distant from cancer cells; a more specific definition of how "proximal" and "distal" are defined would be useful. In addition, there are two potential issues to address or at least discuss: 1) the 2D nature of a tissue section makes it impossible to determine whether cancer cells are present in the same area but above or below the plane of the section. 2) the samples are described as biopsies of resected tumors: the authors should discuss whether the biopsy is likely to reflect the whole tumor or discuss the potential confounding effect of tumor heterogeneity.

Thank you. We have expanded the Materials and methods section to include a more detailed description of how proximal and distal regions are defined. Point 1 has now been addressed in the Discussion. Point 2 has been addressed by further emphasising the use of multiple regions of interest per domain across the tissue section in the *Material and Method*s section. Furthermore, we have also added this point to the Discussion as a limitation of the study.

2) Some of the markers of the distal stroma are also expressed by other cell types – for instance, complement genes are associated with macrophages. In order to confirm that they are indeed expressed in fibroblasts, the authors could consider co-immunofluorescent staining with lineage markers for fibroblasts and myeloid cells, respectively.

Thank you for this advice. We have now considerably extended the transcriptional analysis and this further supports a fibroblast origin. However, we agree with the reviewer that this is not definitive evidence of co-expression. It is possible that technologies with enhanced resolution, such as the CosMx system, may be able to overcome this. We were not able to undertake co-immunofluorescence studies but have added this to the text as an important ambition for future studies.

It is not clear what the take home of Figure 4F is; what kind of differences in transcription factor signatures distinguish the different fibroblast populations is not clear.

We thank the reviewer for this comment. Figure 4F highlights the main functional and TF regulon differences between sc-proximal and sc-distant cells. This informs the likely dominant transcription factors governing the characterised differential gene expression profiles and functional activities of sc-proximal vs sc-distant fibroblasts. We have expanded the text in this section to clarify.

3) Figure 5 seems somewhat disjointed from the rest of the manuscript – it is unclear whether gene signatures that predict poor outcomes are linked to a specific geographic distribution of different types of fibroblasts, or whether it is simply the prevalence of each type that affects tumor characteristics.

Thank you. Figure 5 is included here as it identifies genes associated with survival outcome and interrogates the likely spatially-defined source of such signals. Firstly the-prognosis associated genes are identified by differential expression analysis including all region data stratified by good (3yr) and bad (1yr) survival (Figure 5A). The likely spatially-defined source of prognostic gene signal is then interrogated by distinct analysis comparing good vs bad survival groupings using Immune region or stroma region data only. Any gene found to be significantly divergent (prognosis-associated) from either analysis is visualised in Figure 5C to assess the relative Immune-region and Stromal-region contribution. To further interrogate the source of stromal prognostic gene signals, again, distinct good vs bad differentially expressed genes were identified using either only tumor-proximal or tumor-distant region data (Figure 5D). To summarise the spatially-defined region-specific and non-region specific prognosis associated genes, the overlap of any within PS, DS or Immune region significant hit was visualised in Figure 5E. We have added further text in the manuscript to clarify the inclusion of these analyses.

4) The authors should discuss their findings in light of recent literature that either supports or contradicts the link between myCAFS and outcome, as this is an area of active investigation.

Thank you. We have now extended the discussion considerably and refer the reader to an excellent recent review on PDAC in CAF.

5) In Figure 2B, the colors of the tumor and proximal stroma are difficult to distinguish from one another.

Thank you, colours have now been changed to make them more distinguishable.

6) Figure 2 (and other figures as well) many of the graphs and text is too small to clearly read. For the heat map, one option would be to highlight key genes labelling them with a larger font. The whole list could be moved to supplemental information. Some of the other graphs could be enlarged.

We thank the reviewer for this comment. Figures have now been edited to increase graph and text sizes where possible such that minimum text size is 6pt.